# Prevalence of canine hip dysplasia in 10 breeds in France, a retrospective study of the 1997-2017 radiographic screening period

Arnaud Baldinger[1,2]◉*, Jean-Pierre Genevois[1,2]◉, Pierre Moissonnier[1,2], Anthony Barthélemy[3], Claude Carozzo[1,2], Éric Viguier[1,2], Thibaut Cachon[1,2]◉

1 Surgery Unit, Lyon Veterinary Teaching Hospital, Université de Lyon, VetAgro Sup, Marcy l'Etoile, France,
2 Research Unit ICE, UPSP A104-2016, Université de Lyon, VetAgro Sup, Marcy l'Etoile, France,
3 Intensive Care Unit (SIAMU), Université de Lyon, VetAgro Sup, Marcy l'Etoile, APCSe, France

◉ These authors contributed equally to this work.
* Arnaud.baldinger@vetagro-sup.fr

## Abstract

Canine hip dysplasia (HD) is a complex developmental disease of the coxo-femoral joint and is one of the most common orthopedic conditions in dogs. Due to the genetic contribution, most of the programs fighting against HD recommend selective breeding that excludes affected dogs. Using the best-scoring dogs for breeding may reduce the prevalence of HD. In France, the phenotypic screening of coxo-femoral joint conformation remains a strategy for breeders to establish selection decisions. The HD prevalence was evaluated in 10 breeds, based on the assessment of 27,710 dogs, during the 1997–2017 screening period, which was divided into 3 homogeneous cohorts for analysis. The global HD prevalence varied widely among breeds from 5% (Siberian Husky) to 51.9% (Cane Corso). It decreased over time in 6 breeds, among which 4 (Cane Corso, Gordon Setter, Rottweiler and White Swiss Shepherd) showed a significant decrease. A statistically significant increase in HD prevalence was noted for the Siberian Husky. Although the efficacy of phenotype-based breeding programs remains controversial, our results are in accordance with several recent studies showing that long-term selection policies are valuable, as they may help decreasing the HD prevalence in some breeds. The complementary use of more recent tools such as estimated breeding values and genomics would probably help breeders achieve more substantive results.

## Introduction

Canine hip dysplasia (HD) is a complex developmental disease of the coxo-femoral joint and is one of the most common orthopedic conditions in dogs [1].

Osteoarthritis results from the varying degree of hip laxity of the coxo-femoral joint leading to subluxation of the femoral head, a shallow acetabulum and flattening of the femoral head [2]. These anatomical abnormalities lead to an inflammatory reaction with secondary degenerative joint disease associated with pain and lameness ranging from mild to severe [1,3]. The multifactorial characteristics of this medical condition are the results of genetic and environmental contributions (nutrition, growth rate, overall body weight). Studies have shown that the degree of heritability varies from 0.11 to 0.68 among different dog breeds [4].

**Funding:** The author received no specific funding for this work.

**Competing interests:** The authors have declared that no competing interests exist.

Due to the genetic predisposition, excluding affected dogs from breeding has been shown to reduce the prevalence of HD [5]. The heritability of HD and the response to selection is however breed dependent. The higher the heritability, the greater is the expected genetic improvement over time when selective breeding is practiced [6].

In France, a program intending to reduce HD prevalence was introduced in 1971 under the responsibility of each breed club and supervised by the French Kennel Club (SCC: Société Centrale Canine) [7]. Currently, phenotypic screening for coxo-femoral joint conformation remains a strategy for breeders for making selection decisions [8]. Radiographic screening for HD is based on a conventional ventrodorsal hip extended radiograph in anaesthetized or deeply sedated dogs. According to the Fédération Cynologique Internationale (FCI), a five-class system (A: no signs of HD; B: near normal hip joints, C: mild HD, D: moderate HD, E: severe HD) is used in continental Europe, Asia, Russia and parts of South America. The grades are defined descriptively based on the size of the Norberg angle (NA), depth of the acetabulum, degree of subluxation and signs of secondary joint disease [9].

The minimum age for official screening is 12 months, except in large and giant breeds, in which it is 18 months [7]. For each breed, an official reader is appointed by the breed club to analyse the radiographs. Several control programs have been conducted in different countries over the last 25 years, and their ability to reduce hip dysplasia has shown variable results. While several reports identified a decrease in HD prevalence [5, 7,10,11,12,13,14,15,16], others failed to identify any significant progress [17,18,19,20].

The aim of our observational study was to compare the HD prevalence in affected breeds over several periods of time to evaluate the effectiveness of the hip dysplasia control program in France.

## Materials and methods

### Data

From 1997 to 2017, 40,521 standard radiographs of extended hindlimbs submitted by breeders or owners in 195 different breeds were evaluated independently by the same examiner (JPG) for HD assessment. All the data were computerized and recorded in a single electronic data-base to allow further analysis. For each breed, the incidence of each of the 5 scoring classes was extracted from the database for each year covered in this retrospective study. Breeds were excluded if the creation of 3 homogeneous cohorts of 7 years was not possible or if the total number of radiographs read per breed and per period was insufficient (i.e. <200). Breeds selected were analysed without exclusion nor selection in the database. The same single panel-ist evaluated all included breeds from 1997 to 2017.

### Scoring protocol

All dogs were scored according to the FCI 5 class grading scale. Each joint was assigned to one of five grades (A-E) that are defined descriptively; the final grade refers to the worst joint. A and B are considered as normal joints (non-dysplastic), and grades C, D and E represent mild, moderate and severe dysplasia, respectively. To evaluate and compare the HD prevalence over time, each breed was divided into 3 homogeneous cohorts of 7 years (1997-2003/ 2004-2010/ 2011–2017).

### Statistical analysis

For each breed and each period, HD prevalence (expressed as %) was obtained by dividing the number of dogs that scored C-D and E by the total number of dogs evaluated for the breed.

Within each breed, prevalences among A+B dogs and C+D+E dogs, and between A+B dogs and C, D, E dogs were compared for each period using Fisher's exact test. Statistical analyses were performed by one author (AB) using a commercial software program (Prism 6, Graph-Pad Software, La Jolla, USA, CA).

## Results

Overall, 27,710 records for 10 breeds were included in the study: Alaskan Malamute, Australian Shepherd dog, Berger de Brie, Belgian Shepherd dog, Cane Corso, English Cocker Spaniel, Gordon Setter, Rottweiler, Siberian Husky and White Swiss Shepherd dog. The number of evaluated dogs for the 10 breeds for each study period is presented in Table 1.

The overall prevalence of HD was evaluated in each breed mentioned above. The HD prevalence for each breed varied from 5% (Siberian Husky) to 51.9% (Cane Corso).

A diminishing prevalence of HD was noted in 6 breeds in this study. Between 1997 and 2017, the HD prevalence dropped from 23% to 20.7% in the Berger de Brie, from 72.7% to 49.9% in the Cane Corso, from 23.4% to 18.5% in the English Cocker Spaniel from 36.9% to 23% in the Gordon Setter, from 23.9% to 17% in the Rottweiler and from 34.6% to 20.3% in the White Swiss Shepherd dog. Among all breeds, the largest improvements in the prevalence of HD were noted in the Cane Corso. A significant decrease in HD prevalence (Table 2) was observed in 4 breeds: Cane Corso, Gordon Setter, Rottweiler and White Swiss Shepherd. A non-significant decrease in HD prevalence was reported in 2 other breeds (Berger de Brie and English Cocker Spaniel).

In 5 of the 6 breeds with a decrease in HD prevalence, a marked decrease in the D-E grades was noted, except for the White Swiss Shepherd Dog in which the C and D grades decreased while the E grade remained stable. This was statistically significant for the Cane Corso, the Gordon Setter, the Rottweiler and for the White Swiss Shepherd dog (Table 2).

For the English Cocker Spaniel, a decrease in the C grade in association with the D and E grades was noted but not statistically significant (Table 2).

Three breeds (Australian Shepherd dog, Alaskan Malamute and Belgian Shepherd dog) showed an initial decrease in HD prevalence (13.4% to 11.2%, 13.1% to 10%, 8.1% to 7.8%, respectively) followed by an increase in HD prevalence (11.2% to 13.4%, 10 to 13%, 7.8% to 9.9%, respectively) during the last part of the evaluation, but was not statistically significant over the period of study (Table 2). The Alaskan Malamute showed a stable prevalence of HD over the study period (13.1% to 13%).

A statistically significant increase in HD prevalence (2.9% to 6%) was observed in the Siberian Husky (Table 2).

**Table 1. Number of evaluated dogs (N) for the 10 breeds for each study period.**

| Breed | 1997–2017 (N) | 1997–2003 (N) | 2004–2010 (N) | 2011–2017 (N) |
|---|---|---|---|---|
| Cane Corso | 1338 | 201 | 542 | 595 |
| Gordon Setter | 1803 | 900 | 594 | 309 |
| White Swiss Shepherd dog | 2924 | 225 | 1063 | 1636 |
| Berger de Brie | 1631 | 777 | 573 | 281 |
| Rottweiler | 7072 | 4539 | 1418 | 1115 |
| English Cocker Spaniel | 812 | 203 | 231 | 378 |
| Australian Shepherd dog | 4442 | 210 | 1469 | 2763 |
| Alaskan Malamute | 897 | 206 | 293 | 398 |
| Belgian Shepherd dog | 4998 | 1668 | 1796 | 1534 |
| Siberian Husky | 1870 | 380 | 397 | 1093 |

**Table 2. HD prevalence over several periods of time compared within each period using Fisher's exact test (*p* <0.05).** Bold italic results are statistically significant. A+-B = nondysplastic, C+D+E = dysplastic.

| Breed | Grade scale | Period 1 | | Period 2 | | Period 3 | |
|---|---|---|---|---|---|---|---|
| | | Mean (%) | *p (1 vs. 2)* | Mean (%) | *p (2 vs. 3)* | Mean (%) | *p (1 vs. 3)* |
| **Cane Corso** | A+B | 27.3 | **< 0.0001** | 47.4 | 0.3736 | 50.1 | **< 0.0001** |
| | C+D+E | 72.7 | | 52.6 | | 49.9 | |
| | A+B | 27.3 | **< 0.0001** | 47.4 | 0.1625 | 50.1 | **< 0.0001** |
| | C | 24.3 | | 21.9 | | 24.4 | |
| | D | 24.2 | | 20.4 | | 18.5 | |
| | E | 24.2 | | 10.3 | | 7 | |
| **Gordon Setter** | A+B | 63.1 | **< 0.0001** | 75.4 | 0.6229 | 77 | **< 0.0001** |
| | C+D+E | 36.9 | | 24.6 | | 23 | |
| | A+B | 63.1 | **< 0.0001** | 75.4 | 0.6103 | 77 | **< 0.0001** |
| | C | 19.4 | | 12.1 | | 13.3 | |
| | D | 14.3 | | 9.5 | | 7.8 | |
| | E | 3.2 | | 3 | | 1.9 | |
| **Berger De Brie** | A+B | 77 | 0.5093 | 78.5 | 0.8586 | 79.3 | 0.4526 |
| | C+D+E | 23 | | 21.5 | | 20.7 | |
| | A+B | 77 | 0.0841 | 78.5 | 0.4157 | 79.3 | 0.1855 |
| | C | 13.6 | | 12.4 | | 14.6 | |
| | D | 7.3 | | 8.6 | | 5.7 | |
| | E | 2.1 | | 0.5 | | 0.4 | |
| **White Swiss Shepherd dog** | A+B | 65.4 | **0.002** | 75.5 | **0.0116** | 79.7 | **< 0.0001** |
| | C+D+E | 34.6 | | 24.5 | | 20.3 | |
| | A+B | 65.4 | **0.0103** | 75.5 | 0.0591 | 79.7 | **< 0.0001** |
| | C | 26.4 | | 18 | | 15.6 | |
| | D | 7.7 | | 5.5 | | 4 | |
| | E | 0.5 | | 1 | | 0.7 | |
| **Rottweiler** | A+B | 76.1 | **0.003** | 79.9 | 0.0515 | 83 | **< 0.0001** |
| | C+D+E | 23.9 | | 20.1 | | 17 | |
| | A+B | 76.1 | **0.0288** | 79.9 | **0.0354** | 83 | **< 0.0001** |
| | C | 11.9 | | 9.7 | | 9.9 | |
| | D | 9.7 | | 8.5 | | 5.6 | |
| | E | 2.3 | | 1.9 | | 1.5 | |
| **English Cocker Spaniel** | A+B | 76.6 | 0.6464 | 78.3 | 0.347 | 81.5 | 0.1599 |
| | C+D+E | 23.4 | | 21.7 | | 18.5 | |
| | A+B | 78.6 | 0.8883 | 78.3 | 0.6953 | 81.5 | 0.2524 |
| | C | 16.7 | | 16.5 | | 15.1 | |
| | D | 4.7 | | 3.5 | | 2.4 | |
| | E | 2 | | 1.7 | | 1 | |
| **Australian Shepherd dog** | A+B | 86.6 | 0.3565 | 88.8 | **0.0464** | 86.6 | >0.9999 |
| | C+D+E | 13.4 | | 11.2 | | 13.4 | |
| | A+B | 86.6 | **0.0075** | 88.8 | **0.0083** | 86.6 | **< 0.0001** |
| | C | 6.7 | | 8.3 | | 11.2 | |
| | D | 6.7 | | 2.5 | | 1.7 | |
| | E | 0 | | 0.4 | | 0.5 | |

(*Continued*)

**Table 2.** (Continued)

| Breed | Grade scale | Period 1 | | Period 2 | | Period 3 | |
|---|---|---|---|---|---|---|---|
| | | Mean (%) | p (1 vs. 2) | Mean (%) | p (2 vs. 3) | Mean (%) | p (1 vs. 3) |
| **Alaskan Malamute** | A+B | 86.9 | 0.3134 | 90 | 0.2318 | 87 | >0.9999 |
| | C+D+E | 13.1 | | 10 | | 13 | |
| | A+B | 86.9 | 0.1737 | 90 | 0.1914 | 87 | 0.9599 |
| | C | 6.3 | | 7.3 | | 6.6 | |
| | D | 6.3 | | 2.4 | | 5.6 | |
| | E | 0.5 | | 0.3 | | 0.8 | |
| **Belgian Shepherd dog** | A+B | 91.9 | 0.3246 | 92.2 | **0.0316** | 90.1 | 0.0724 |
| | C+D+E | 8.1 | | 7.8 | | 9.9 | |
| | A+B | 91.9 | 0.3387 | 92.2 | **0.0392** | 90.1 | **0.0025** |
| | C | 5.2 | | 5.8 | | 8 | |
| | D | 2.4 | | 1.7 | | 1.4 | |
| | E | 0.5 | | 0.3 | | 0.5 | |
| **Siberian Husky** | A+B | 97.1 | 0.2996 | 95.7 | 0.1913 | 94 | **0.0177** |
| | C+D+E | 2.9 | | 4.3 | | 6 | |
| | A+B | 97.1 | 0.2849 | 95.7 | 0.5857 | 94 | 0.0524 |
| | C | 2.6 | | 3.8 | | 5.3 | |
| | D | 0 | | 0.5 | | 0.6 | |
| | E | 0.3 | | 0 | | 0.1 | |

In 2 of the 3 breeds with an increased HD prevalence between the first and the third period of study, there was an increase in the C grade associated with a decrease in D grade (Australian Shepherd Dog and Belgian Shepherd dog) while the E grade remained stable. These observations were statistically significant. For the Siberian Husky, an increase of the C and D grades was noted, although not statistically significant.

The prevalence of HD over the different periods of time is shown in Figs 1 and 2.

## Discussion

A diminishing prevalence of HD was noted in 6 breeds in this study. Among them, 4 breeds (Cane Corso, Gordon Setter, Rottweiler and White Swiss Shepherd) showed a significant change in HD prevalence over the study period. These results support the fact that a long-term purely phenotypic selection mode against hip dysplasia based on radiographic screening control might be efficient in decreasing the HD prevalence.

The increase in the C grade noted in 3 breeds (Australian Shepherd dog, Belgian Shepherd Dog and Siberian Husky) with an increased HD prevalence is difficult to explain, and the situation is most likely different from breed to breed. We may consider that, for a while, the selection was potentially not strong enough in some breeds. We could also assume that, for the Australian Shepherd, for instance, the increase in the B grade led to an increase in B to B mating (instead of A to A or A to B mating), which, due to the genetic recombination, could result in an increased risk of obtaining C scoring dogs in the offspring. However, the variation between the initial and final period in terms of HD prevalence noted in these breeds was less than 3.1%. This increase among the initial and final period remained slight and not significant, except for the Siberian Husky.

A previous study demonstrated that when all dogs in a breed have nearly the same hip phenotype, almost no selection pressure can be applied to improve hip quality based on hip radiograph screening [14]. According to the results of the present study, this was potentially the

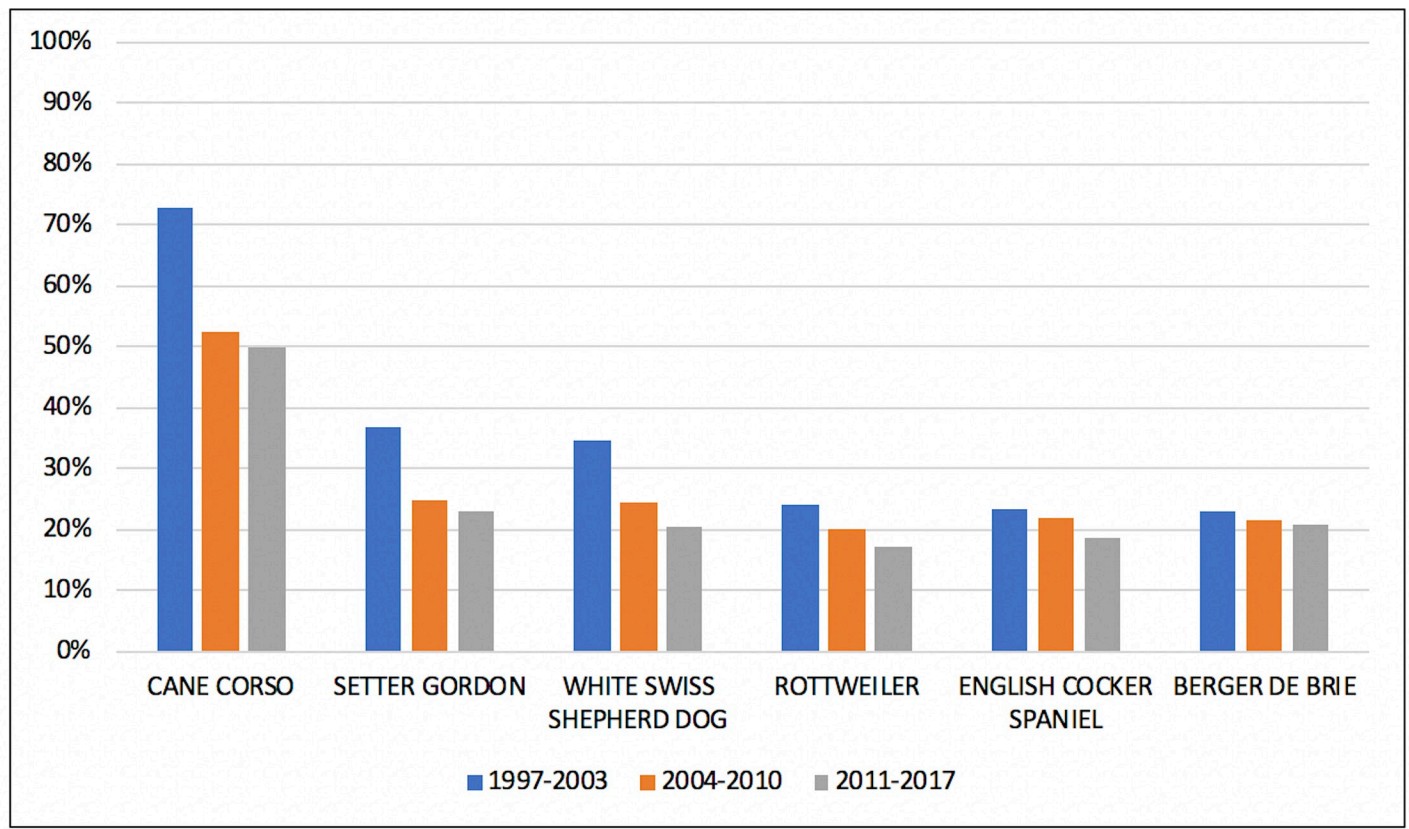

**Fig 1. Diminishing prevalence of HD in the Cane Corso, Gordon Setter, White Swiss Shepherd dog, Rottweiler, English Cocker Spaniel and Berger de Brie from 1997 to 2017.**

case for the Siberian Husky and for the Australian Shepherd dog, which demonstrated slight changes in HD prevalence. This was also potentially the case for the Cane Corso, the Gordon Setter and the Rottweiler between the second and the third period of the study where the diminishing prevalence of HD was not statistically significant.

Overall, 9 breeds had a prevalence of moderate and severe HD (D-E grades) lower than 10% which is consistent with the results of a recent survey [21]. The Cane Corso had a prevalence of D-E grades lower than 25%. There is still a margin for improvement in this breed, although it showed the largest improvements in the prevalence of HD. These results are consistent with previous studies indicating that selective breeding using classifications of hip joint phenotypes might improve hip conformation in several breeds of dogs [5,7,10,11,12,13,14,15,16], although other studies showed different findings, and the efficiency of using screening programs to reduce the prevalence of HD has been questioned [17,18,19,20].

These results must be interpreted with caution since the evaluation of coxo-femoral joint status is not mandatory for breeding in France [7]. In a 1993–2002 survey [22], it was demonstrated that in France, only 2 to 19% of the dogs were screened for HD. Although the number of screened dogs has increased since this period, it is likely that, depending on the breed, a small fraction of all breeding dogs undergo a hip radiograph. Moreover, there is an unknown proportion of veterinarian (or owner) prescreening of the radiographs with obvious hip dysplasia, leading to the lack of presentation of the "worst" radiographs for official screening. Therefore, our data reflect only those dogs whose owners and breeders submitted radiographs

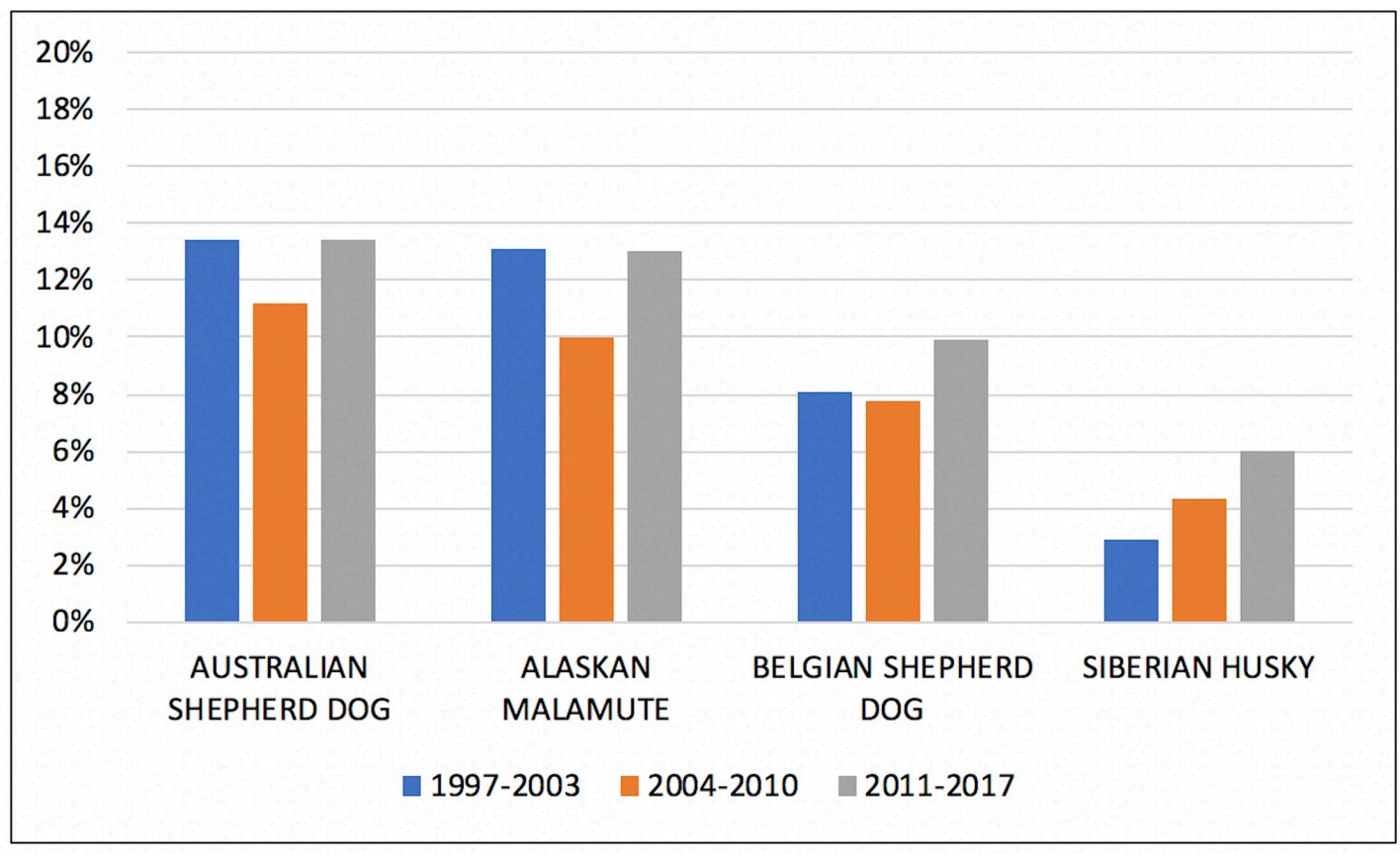

**Fig 2. Diminishing and increasing prevalence of HD in the Australian Shepherd dog, Alaskan Malamute, and Belgian Shepherd dog; increasing prevalence of HD in the Siberian Husky; from 1997 to 2017.**

for analysis. This proportion varies from 20 to 40% in Switzerland [23]. In Scandinavian countries, all breeding animals in control-program breeds are screened, and both dam and sire hip radiographs need to be submitted for screening to be registered by the Kennel Club [13,17,18]. In these countries, an HD control program gives a good overall prevalence for each breed. A study demonstrated that an improvement in hip quality can be achieved by selection based on the subjective scoring of radiographs when all dogs of a breed are evaluated [14].

Therefore, the lack of breeding restrictions in France and other countries [5] (United Kingdom, United States of America) and the associated lower scoring rate might explain the smaller degree of progress for some breeds. As previously mentioned, the true prevalence of HD could be higher than that depicted by our results because they reflect only the results of the radiographs submitted for official screening [7].

Most of the French breed clubs that are involved in a HD control program encourage breeders to have their breeding stock and offspring radiographed through a scoring grid which takes into account the fact that the dog and/or some of its offspring have been submitted to radiographic hip scoring and the results of the scoring. Every result from an official hip scoring that is communicated by the breed club to the French Kennel Club (SCC) is mentioned on the dog's pedigree, which is now a 5-generation pedigree document. The result is also registered on an open access internet portal created by the SCC named LOF Select. This portal enables breeders to access the characteristics of every registered dog, look for a breeding dog, and create virtual matings. Furthermore, the SCC is involved in a program to calculate and set up

estimated breeding values, to help breeders choose their breeding stock. A project to create a reproductive ability certification, which would involve health characteristics (including hip status) is under study. It would undoubtedly increase the number of radiographed dogs.

The HD screening system is based on a subjective evaluation of radiographic findings. Panelist dependent variation is possible, and it has been demonstrated that significant intra- and inter-observer variation in classification may occur [24]. In our study, all breeds were evaluated by the same single panelist, which avoided interobserver variability, yet an intraobserver variability over the long study period cannot be totally excluded which could introduce bias.

In our study, multiple anesthesia/sedation protocols were used. No standardized protocols have been proposed for performing hip radiographs, as it has been decided that, for safety reasons, the best protocol is the one the veterinarian is comfortable using. A Scandinavian study [25] showed that acepromazine should not be used for sedation because it causes a very poor myoresolution. A study concerning the type of chemical restraint used by French veterinarians performing HD screening radiographs [26,27] showed that these protocols (mostly a single injection of α-2 agonist, or an association between α-2 agonist and other injectable drugs such as diazepam, ketamine or butorphanol) are acceptable based on the FCI standard requirements for HD screening.

A study demonstrated a strong association between the radiographic scoring of hip status and subsequent incidence of veterinary care and mortality related to HD in five breeds of dogs. It demonstrated that the selection of breeding stock based on the screening results with regard to hip status can be expected to reduce the risk of clinical problems related to HD [28] which also emphasizes the interest in and effect of selection based on hip radiograph screening to reduce the HD prevalence.

Several other approaches to assessing the coxo-femoral joint status have been proposed, such as hip joint laxity measurements (distraction methods, distraction Norberg angle) and the use of estimated breeding values.

Distraction methods were first described by the PennHip organization [29] and have been shown to be reliable screening methods for predicting hip joint degeneration [30]. A recent study [31] evaluated the correlation between the distraction angle (DI) and the distraction Norberg angle measured at 4 months of age, and the official FCI hip score determined at 12 months of age. It was shown that the distraction Norberg angle had a fair correlation with the DI at 4 months and therefore reflects hip passive laxity. It also demonstrated that 98% of hips with a distraction Norberg angle higher than 85˚ at 4 months had an A, B or C FCI score at 12 months of age.

To reduce the incidence of HD, many researchers have recommended the use of estimated breeding values (EBV) to improve the rate of genetic progress in terms of selection against HD [17,32,33,35,36].

A study showed that the EBV is more accurate and abundant than the phenotype [32] and provides more reliable information on the genetic risk of disease for a greater proportion of the population. An efficient selection mode is to include information about the hip status of relatives because the inheritance of HD is still unclear, and dogs with phenotypic normal hip joints may carry genes leading to HD in their offspring [34]. A recent study confirmed that using phenotypic health information and selecting sires and dams from pedigrees free from HD improves hip joint health and therefore reduces the HD prevalence [35,37].

There is a great deal of research based on genomics and DNA testing related to canine HD [38,39,40,41], some of which is linked to similar human pathology [42]. It is beyond the scope of this study to address this very specific research area, but it is likely that, in the foreseeable future, new tools will complement radiographic examination of the coxo-femoral joint in order to prevent canine HD.

## Conclusions

This study confirms that long-term selection based on hip radiograph screening reduced the HD prevalence from 1997 to 2017 in the Cane Corso, Gordon Setter, Rottweiler and White Swiss Shepherd. It demonstrated that phenotypic selection for hip conformation may be effective and should be continued, although it is dependent on the voluntary participation of breeders and owners. Some breeds demonstrated slight changes in HD prevalence, however, when breeds have nearly the same hip phenotype, almost no selection pressure can be applied to improve hip quality based on hip radiograph screening. The true prevalence of HD in the breeds presented in this study is probably higher than those reported in our results. However, this screening type remains the only official procedure in most countries. To achieve a further decrease in the HD prevalence, communicating with veterinary practitioners and breeders on the value of classification is necessary in association with the use of EBV and genomic selection which should be considered.

## Acknowledgments

The authors want to thank Dr. Thomas Lecoq for his assistance with this manuscript.

## Author Contributions

**Conceptualization:** Arnaud Baldinger, Jean-Pierre Genevois, Thibaut Cachon.

**Formal analysis:** Anthony Barthélemy.

**Supervision:** Jean-Pierre Genevois, Thibaut Cachon.

**Validation:** Jean-Pierre Genevois, Thibaut Cachon.

**Visualization:** Arnaud Baldinger, Jean-Pierre Genevois, Thibaut Cachon.

**Writing – original draft:** Arnaud Baldinger, Jean-Pierre Genevois, Thibaut Cachon.

**Writing – review & editing:** Jean-Pierre Genevois, Pierre Moissonnier, Anthony Barthélemy, Claude Carozzo, Éric Viguier, Thibaut Cachon.

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
