## [Decision Letter · Decision Letter 0]

30 Apr 2020

PONE-D-20-10280

Prevalence of canine hip dysplasia in 28 breeds in France, a retrospective study of the 1993-2019 radiographic screening period

PLOS ONE

Dear Dr Baldinger

Thank you for submitting your manuscript to PLOS ONE. After careful consideration, we feel that it has merit but does not fully meet PLOS ONE’s publication criteria as it currently stands. Therefore, we invite you to submit a revised version of the manuscript that addresses the points raised during the review process.

Many thanks for submitting your manuscript to PLOS One

Your manuscript was reviewed by three experts in the field. One recommended rejection, while two recommended major revision.

I have decided to give you the opportunity to revise the manuscript in line with the reviewers comments, and then resubmit it.

The manuscript will be returned (where possible) to the same reviewers.

If you could write a response to reviewers comments that will help expedite review when it is resubmitted.

Wishing you the best of luck with your revisions

Hope you are keeping safe and well in these difficult times

Thanks

Simon

We would appreciate receiving your revised manuscript by Jun 14 2020 11:59PM. To enhance the reproducibility of your results, we recommend that if applicable you deposit your laboratory protocols in protocols.io, where a protocol can be assigned its own identifier (DOI) such that it can be cited independently in the future. For instructions see: http://journals.plos.org/plosone/s/submission-guidelines#loc-laboratory-protocols

We look forward to receiving your revised manuscript.

Kind regards,

Simon Russell Clegg, PhD

Academic Editor

PLOS ONE

Reviewers' comments:

Reviewer's Responses to Questions

**Comments to the Author**

1. Is the manuscript technically sound, and do the data support the conclusions?

Reviewer #1: No

Reviewer #2: Partly

Reviewer #3: Partly

2. Has the statistical analysis been performed appropriately and rigorously? 

Reviewer #1: No

Reviewer #2: No

Reviewer #3: No

3. Have the authors made all data underlying the findings in their manuscript fully available?

Reviewer #1: No

Reviewer #2: Yes

Reviewer #3: Yes

4. Is the manuscript presented in an intelligible fashion and written in standard English?

Reviewer #1: No

Reviewer #2: Yes

Reviewer #3: Yes

5. Review Comments to the Author

Reviewer #1: I think it is an interesting manuscript with important new information about the degree of hip dysplasia/control program in France in some dog breeds. However I have some important doubts used that I would like to clarify:

I think that the presented methodology should improve: the data are “From 1993 to 2019, 50955 standard radiographs of extended hindlimbs submitted by breeders or

91 owners were evaluated independently by the same examiner (JPG) for HD assessment”

How many radiographs on database were not included in each breed? This selection of data present important Bias that can change the results. So, I suggest also to change the title the manuscript

The paper have a lot of tables and figures Table 1 can be deleted and the information added in text.

Lines 94-97 rewrite (is confusing I don`t understand the main ideas)

Lines 103-105 “… each breed was divided into 2 or 3 several cohorts, depending on the length of their screening period.” I do not understand the criteria used to avoid any kind of Bias, should be used fixed periods of time for all breeds (eg: 5 years) and if there are no animals in the breed they are left blank.

Delete Table 2 and 3– redundant information

Delete Figure1 and Table 4 – I think that the global prevalence o HD in breeds include animals from 1993 and 2019 to study the prevalence has little interest, the important thing is to have the prevalence per year / limited set of years and to analyze the evolution.

The discussion was not evaluated as I think that the results can be Biased due to the used methodology

Reviewer #2: In the manuscript entitled “Prevalence of canine hip dysplasia in 28 breeds in France, a retrospective study of the 1993-2019 radiographic screening period” the authors present data on a large number of dog breeds evaluated for hip dysplasia in France. The authors conclude that their study essentially replicates a large number of published studies in that phenotypic selection may have reduced the incidence of hip dysplasia of some breeds over time. The authors should be commended on noting that the data were biased because of the voluntary nature of the screening scheme in France.

Essentially, the authors presented data over time for a number of dog breeds with very little additional insight than what is already in the literature. Replication of previous findings and reports in another country with a different set of examiners appears to be the contribution of this manuscript.

Importantly, the presentation of the data needs revisiting. There is no synthesis of the authors findings as evidenced by 28 individual figures, one for each breed.

Furthermore, there are some methodological questions that need to be addressed:

1. The authors state that a breed club appoints the radiographic reader for a given breed. That would imply there is no oversight or continuity across breeds and improvement of lack thereof in hip conformation can be a reflection of a single individual’s subjective view (line 75). That would introduce distinct bias. The authors appear to have tried to interject consistency though in that if the reader had not read for the entire 26/27 year period, the data appear to have been truncated to only the time period of one reader (lines 142-143). This is contradicted by lines 94-95 which state the breeds were excluded if the reader changed. And then on line 91, the authors state that one author (JPG) evaluated 50,955 radiographs independently (which is in excess of the number of dogs (47,895) the authors state are in the study on line 30. The discussion states all breeds were evaluated by the same examiner (lines 335-336). This leads to the question of whose rating was used in the study? If it was JPG’s then why exclude breeds? Did JPG randomize evaluations or do breed by breed?

a. The authors need to clarify what readings were used in the study

b. and also note the bias within a breed due to one reader doing all assessments and how that could play a role in differential breed selection responses.

2. The statistical analyses were done on the combined data to create binomial yes hip dysplasia, no hip dysplasia, yet the authors discuss the individual grades of dysplastic hips (e.g., C, D, and E) which were not analyzed by their statistical model (as just one example of this, line 267)

3. Table 3 gives the time periods used in the analyses. These periods appear arbitrary with no explanation given as to why these time frames were chosen. And the authors then do not take this into account when analyzing. A much more sophisticated analytical model is warranted for the disparate (and multiple) time periods than what the authors have done in order to really gain insight into the progress of using radiographic screening.

4. Table 4 appears to have presented the breeds in order of hip dysplasia prevalence, and presents number and prevalence for 32 breed categories. One breed category (Belgian Shepherd dog) was particularly noticeable because of the multiple entries for each variety and then presumably a composite entry? What was the rationale for that? This also needs to be stated explicitly that the authors elected to present in this way.

Lines 257-271 in the discussion are all results and should be in the results section.

Line 339 “for security reasons” is not correct

Line 343 “showed that these protocols” it is unclear to what protocols “these protocols” are referring.

Line 351 -353. The authors state that it is “unlikely that the HD prevalence can be reduced much further based only on the radiographic screening control.” What are the authors basing that statement upon? Their own data show that the prevalence continued to decline in those breeds making progress and many published studies show a continued downward trend. Furthermore the authors state that a relatively low proportion of dogs are radiographed.

The use of the abbreviation of DNA for the distraction Norberg angle should be replaced as DNA has a universal genetic meaning –this is especially important when the authors introduce genomic concepts juxtaposed to the Norberg angle abbreviation in line 373.

The conclusion adds nothing to the literature.

Reviewer #3: This is a generally well written paper and the topic is interesting.

Nonetheless, I have a few concerns regarding the handling of the data:

1. The time frame of the observations vary greatly among breeds and, therefore, they are poorly comparable. Indeed, some breeds have data covering more than 30 ears and some others only 6 or 7 years. I see that the authors have chosen strict exclusion criteria to limit inter-observer variability in the evaluation of HD, and I also see that such a limitation is difficult to overcome. Therefore I think that the authors should discuss broadly such differences in the discussion.

2. At lines 267-276 the authors describe the variations of the different HD grades in time. I think that this part belongs to the results section. Moreover, I do not see any reference to such a comparison in the description of the statistical analysis. The authors should run appropriate statistical tests on this part (that is, in my opinion, the most interesting). A simple description of the trend in the graphs is not enough to make conclusions.

3. The authors should find a way to summarize the graphical representation of the data. Presenting 28 different graph is very uncomfortable to read.

6. PLOS authors have the option to publish the peer review history of their article (what does this mean?). If published, this will include your full peer review and any attached files.

Reviewer #1: No

Reviewer #2: No

Reviewer #3: No

---

## [Author Response · Author response to Decision Letter 0]

19 May 2020

Response to reviewers and academic editor : 

We would like to thank the reviewers but also the academic editor for their comments and suggestions which contributed to our article improvement. 

Reviewer #1: (changes requested by reviewer 1 are highlighted in blue in the manuscript)

Dear Reviewer 1, 

We thank you for your interest in our manuscript. We appreciate your constructive comments. Please find hereunder the changes made to the manuscript following your comments. 

I think it is an interesting manuscript with important new information about the degree of hip dysplasia/control program in France in some dog breeds. However I have some important doubts used that I would like to clarify:

I think that the presented methodology should improve: the data are “From 1993 to 2019, 50955 standard radiographs of extended hindlimbs submitted by breeders or owners were evaluated independently by the same examiner (JPG) for HD assessment”

How many radiographs on database were not included in each breed? This selection of data present important bias that can change the results. So, I suggest also to change the title the manuscript

Thank you for this positive comment. According to your comments, we changed the number of breeds evaluated (10 breeds from 1997 to 2017). The database contains the radiographic screening parameters of 40,521 official readings of standard hip extended radiographs performed by JPG, full professor at the Small Animal Department of the Veterinary School of Lyon, and main French panelist for HD/ED/OCD/TLV, JPG is also a member of the scientific commission of the French Kennel Club (Société Centrale Canine), where he is in charge of the French policy against canine hereditary bone and joint diseases. JPG is an expert for the French appeal procedure (in accordance to the 46-2009 FCI circular: « At national level, each FCI member or contract partner should provide an appeal procedure » ), for every radiograph related to a breed club for which he is not the official reader. He is also an expert (all breeds) for the FCI trans-national appeal procedure. As a consequence, the database contains the details of every analysis performed by JPG on an official screening radiograph related to a breed for which JPG has been chosen as an official reader by the breed club. Presently JPG is an official reader for 30 French breed clubs (that is more than 50 breeds as some breeds club are in charge of more than one breed). As it is not compulsory in France to submit hip radiographs to an official scoring, though it is encouraged by the breed-clubs, the number of radiographs sent to official examination is highly variable from one breed to another one (this situation is the same in most of other countries). It has been decided that, to be considered for the study, the minimum number of radiographs per breed should be superior to 200. In France, a breed club may decide to change his official reader whenever he wants. As stated earlier, JPG is presently a reader for 30 French breed clubs, but, since 1997, he had also been a reader for several breed clubs (for instance Akita Inu, Bearded Collie, Bernese Mountain dog, Great dane, Samoyed, Shar pei, Tibetan Dogo). The results of the related hip radiograph analysis are included in the database presented in the materials and methods, but they have not been considered as it has been decided that only the breeds for which JPG was still an official reader in 2017 would be part of the study. These are the 10 breeds included in the study. In each of these breeds, all radiographs (with no exclusion nor selection) which were sent for official examination have been considered. 

The database of 40521 contains also the results of every radiograph examined by JPG for the French Kennel Club appeal procedure. The related breeds are all the breeds for which JPG is not the official reader. The number of radiographs in each of these breeds is variable, but most often it is a limited one. As a consequence, the radiographs which are part of the 40,521 (on December 31, 2017) referenced analysis belong to 195 different breeds. From this database, we extracted all the data related to the 10 breeds (27,710 readings) selected for the study with no exclusion or selection. 

The title of the manuscript was changed accordingly: Prevalence of canine hip dysplasia in 10 breeds in France, a retrospective study of the 1997-2017 radiographic screening period.

The paper have a lot of tables and figures Table 1 can be deleted and the information added in text.

Thank you for this comment. We deleted Table 1 and added the following information in the text. 

Line 66: According to the Fédération Cynologique Internationale (FCI), a five-class system (A: no signs of HD; B: near normal hip joints, C: mild HD, D: moderate HD, E: severe HD) is used in continental Europe, Asia, Russia and parts of South America.

Lines 94-97 rewrite (is confusing I don`t understand the main ideas)

For each breed, the incidence of each of the 5 scoring classes (Table 1) was extracted from the database for each year covered in this retrospective study. Breeds were excluded if the single panelist changed before the end (2019) of the study period, if the total number of radiographs read per breed was insufficient (i.e. <200) and when the radiograph scoring was carried out as part of the French Kennel Club (Société Centrale Canine) appeal procedure.

Thank you for this comment. In order to simplify the exclusion criteria, we removed the part about the French Kennel Club appeal procedure which was confusing and added that breed were analyzed without exclusion nor selection. We also added that the single panelist (JPG) evaluated each breed individually from 1997 to 2017 as the sentence about removing the breed if the single panelist changed before the end of 2017 was also confusing. 

Line 96: For each breed, the incidence of each of the 5 scoring classes was extracted from the database for each year covered in this retrospective study. Breeds were excluded if the creation of 3 homogeneous cohorts of 7 years was not possible, if the total number of radiographs read per breed and per period was insufficient (i.e. <200). Breeds selected were analysed without exclusion nor selection in the database. The single panelist evaluated each breed from 1997 to 2017. 

Lines 103-105 “… each breed was divided into 2 or 3 several cohorts, depending on the length of their screening period.” I do not understand the criteria used to avoid any kind of Bias, should be used fixed periods of time for all breeds (eg: 5 years) and if there are no animals in the breed they are left blank.

Thank you for this constructive comment. Although we were not comparing HD prevalence between breeds, we agree that the division into several heterogenous cohorts is difficult to understand and may introduce bias. As a consequence, we will only keep, for the study, the 10 breeds which were divided in 3 homogeneous cohorts of 7 years from 1997 to 2017 : the Alaskan Malamute, the Australian Shepherd, the Belgian shepherd dog, the Berger de Brie, the Cane Corso, the English Cocker Spaniel, the Gordon Setter, the Rottweiler, the Siberian Husky and the White Shepherd dog.

Line 107: To evaluate and compare the HD prevalence over time, each breed was divided into 3 homogeneous cohorts of 7 years.

Delete Table 2 and 3– redundant information

Thank you for this constructive comment. We deleted table 2 and 3 accordingly.

Delete Figure1 and Table 4 – I think that the global prevalence of HD in breeds include animals from 1993 and 2019 to study the prevalence has little interest, the important thing is to have the prevalence per year / limited set of years and to analyze the evolution.

Thank you for this comment. As we modified the figure with only 10 breeds we chose to keep this figure as we are mentioning global prevalence of HD for the different breeds in the manuscript. We also think that this figure is interesting because for example the Siberian Husky showed a very low prevalence of HD, yet the augmentation of HD prevalence over time is significant. This figure helps us to characterize these variations. 

The discussion was not evaluated as I think that the results can be biased due to the used methodology.

Thank you for this comment. We understand your point of view and hope that our modifications will fulfill your requests. 

Reviewer #2: (changes requested by reviewer 2 are highlighted in red in the manuscript).

Many thanks for the time invested for your constructive comments. Please find hereunder the changes made to the manuscript according to your comments. 

In the manuscript entitled “Prevalence of canine hip dysplasia in 28 breeds in France, a retrospective study of the 1993-2019 radiographic screening period” the authors present data on a large number of dog breeds evaluated for hip dysplasia in France. The authors conclude that their study essentially replicates a large number of published studies in that phenotypic selection may have reduced the incidence of hip dysplasia of some breeds over time. The authors should be commended on noting that the data were biased because of the voluntary nature of the screening scheme in France.

Essentially, the authors presented data over time for a number of dog breeds with very little additional insight than what is already in the literature. Replication of previous findings and reports in another country with a different set of examiners appears to be the contribution of this manuscript.

Thank you so much for your positive comments on our paper. We are grateful for your feedback.

Importantly, the presentation of the data needs revisiting. There is no synthesis of the authors findings as evidenced by 28 individual figures, one for each breed.

Thank you for this comment. In order to synthetize the data, we chose to present our results in 4 figures (one showing the breeds with a diminishing prevalence of HD, one showing the breeds with a diminishing and increasing prevalence of HD over the study period or increasing prevalence of HD. The 2 other figures present the same breeds but with the proportions of A-B grades and C-D-E grades over the study period). We think that the presentation of the results is therefore easier. 

Line 177 to 186: Figures 2 to 5. 

Furthermore, there are some methodological questions that need to be addressed:

1. The authors state that a breed club appoints the radiographic reader for a given breed. That would imply there is no oversight or continuity across breeds and improvement of lack thereof in hip conformation can be a reflection of a single individual’s subjective view. That would introduce distinct bias. The authors appear to have tried to interject consistency though in that if the reader had not read for the entire 26/27 year period, the data appear to have been truncated to only the time period of one reader. This is contradicted by lines 94-95 which state the breeds were excluded if the reader changed. And then on line 91, the authors state that one author (JPG) evaluated 50,955 radiographs independently (which is in excess of the number of dogs (47,895) the authors state are in the study on line 30. The discussion states all breeds were evaluated by the same examiner (lines 335-336). This leads to the question of whose rating was used in the study? If it was JPG’s then why exclude breeds? Did JPG randomize evaluations or do breed by breed?

a. The authors need to clarify what readings were used in the study

b. and also note the bias within a breed due to one reader doing all assessments and how that could play a role in differential breed selection responses.

Thank you for this comment. The database contains the radiographic screening parameters of 40521 official readings of standard hip extended radiographs performed by JPG, full professor at the Small Animal Department of the Veterinary School of Lyon, and main French panelist for HD/ED/OCD/TLV, JPG is also a member of the scientific commission of the French Kennel Club (Société Centrale Canine), where he is in charge of the French policy against canine hereditary bone and joint diseases. JPG is an expert for the French appeal procedure (in accordance to the 46-2009 FCI circular: « At national level, each FCI member or contract partner should provide an appeal procedure » ), for every radiograph related to a breed club for which he is not the official reader. He is also an expert (all breeds) for the FCI trans-national appeal procedure. As a consequence the database contains the details of 1/ every analysis performed by JPG on an official screening radiograph related to a breed for which JPG has been chosen as an official reader by the breed club. Presently JPG is an official reader for 30 French breed clubs (that is more than 50 breeds as some breeds club are in charge of more than one breed). As it is not compulsory in France to submit hip radiographs to an official scoring, though it is encouraged by the breed-clubs, the number of radiographs sent to official examination is highly variable from one breed to another one (this situation is the same in most of other countries). It has been decided that, to be considered for the study, the minimum number of radiographs per breed should be superior to 200. In France, a breed club may decide to change his official reader whenever he wants. As stated earlier, JPG is presently a reader for 30 French breed clubs, but, since 1997, he had also been a reader for several breed clubs (for instance Akita Inu, Bearded Collie, Bernese Mountain dog, Great dane, Samoyed, Shar pei, Tibetan Dogo). The results of the related hip radiograph analysis are included in the database presented in the materials and methods, but they have not been taken into account as it has been decided that only the breeds for which JPG was still an official reader in 2017 would be part of the study. These are the 10 breeds included in the study. In each of these breeds, all the radiographs (with no exclusion nor selection) which were sent for official examination have been taken into account. 

The database of 40521 contains also 2/ the results of every radiograph examined by JPG for the French Kennel Club appeal procedure. The related breeds are all the breeds for which JPG is not the official reader. The number of radiographs in each of these breeds is variable, but most often it is a limited one. As a consequence, the radiographs which are part of the 40521 (on December 31, 2017) referenced analysis belong to 195 different breeds. From this database, we extracted all the data related to the 10 breeds (27710 readings) selected for the study with no exclusion or selection. 

We also noted line 336 that an intraobserver variability over the long study period cannot be totally excluded which could introduce bias.

2. The statistical analyses were done on the combined data to create binomial yes hip dysplasia, no hip dysplasia, yet the authors discuss the individual grades of dysplastic hips (e.g., C, D, and E) which were not analyzed by their statistical model (as just one example of this, line 267)

Thank you for this interesting comment. We modified the statistical analysis accordingly. 

Line 111: For each breed and each period, HD prevalence (expressed as %) was obtained by dividing the number of dogs that scored C-D and E by the total number of dogs evaluated for the breed. 

Within each breed, prevalences among A+B dogs and C+D+E dogs; and between A+B dogs and C, D and E dogs; for each period were compared using Fisher's exact test. Statistical analyses were performed by one author (AB) using a commercial software program (Prism 6, GraphPad Software, La Jolla, USA, CA).

3. Table 3 gives the time periods used in the analyses. These periods appear arbitrary with no explanation given as to why these time frames were chosen. And the authors then do not take this into account when analyzing. A much more sophisticated analytical model is warranted for the disparate (and multiple) time periods than what the authors have done in order to really gain insight into the progress of using radiographic screening.

Thank you for this comment. Although we were not comparing HD prevalence between breeds, we agree that the division into several heterogenous cohorts is difficult to understand and may introduce bias. As a consequence, we will only keep, for the study, the 10 breeds which were divided in 3 homogeneous cohorts of 7 years from 1997 to 2017: the Alaskan Malamute, the Australian Shepherd, the Belgian shepherd dog, the Berger de Brie, the Cane Corso, the English Cocker Spaniel, the Gordon Setter, the Rottweiler, the Siberian Husky and the White Shepherd dog.

4. Table 4 appears to have presented the breeds in order of hip dysplasia prevalence, and presents number and prevalence for 32 breed categories. One breed category (Belgian Shepherd dog) was particularly noticeable because of the multiple entries for each variety and then presumably a composite entry? What was the rationale for that? This also needs to be stated explicitly that the authors elected to present in this way.

Thank you for this comment, you are right, there is no point to study each variety of Belgian Shepherd dog, this has been deleted

Lines 257-271 in the discussion are all results and should be in the results section. 

Thank you for this comment. This section was placed in the results section (Line 147 to 175).

Line 339 “for security reasons” is not correct

Thank you for this comment. This was modified accordingly. 

Line 275: In our study, multiple anesthesia/sedation protocols were used. No standardized protocols have been proposed for performing hip radiographs, as it has been decided that, for safety reasons, the best protocol is the one the veterinarian is comfortable using.

Line 343 “showed that these protocols” it is unclear to what protocols “these protocols” are referring. 

Thank you for this comment. This was modified accordingly. 

Line 279: A study concerning the type of chemical restraint used by French veterinarians performing HD screening radiographs �25,26� showed that these protocols (mostly a single injection of �-2 agonist, or an association between �-2 agonist and other injectable drugs such as diazepam, ketamine or butorphanol) are acceptable based on the FCI standard requirements for HD screening.

Line 351 -353. The authors state that it is “unlikely that the HD prevalence can be reduced much further based only on the radiographic screening control.” What are the authors basing that statement upon? 

Their own data show that the prevalence continued to decline in those breeds making progress and many published studies show a continued downward trend. Furthermore the authors state that a relatively low proportion of dogs are radiographed.

Thank you for this comment, you are right, we deleted this sentence (Line 289). 

The use of the abbreviation of DNA for the distraction Norberg angle should be replaced as DNA has a universal genetic meaning –this is especially important when the authors introduce genomic concepts juxtaposed to the Norberg angle abbreviation in line 373.

Thank you for this comment, we did not use this abbreviation and modified the manuscript accordingly. 

Lines 294: A recent study �30� evaluated the correlation between the distraction angle (DI) and the distraction Norberg angle measured at 4 months of age, and the official FCI hip score determined at 12 months of age. It was shown that the distraction Norberg angle had a fair correlation with the DI at 4 months and therefore reflects hip passive laxity. It also demonstrated that 98% of hips with a distraction Norberg angle higher than 85° at 4 months had an A, B or C FCI score at 12 months of age. D and E FCI scores at 12 months cannot be reliably predicted from the 4-month value of DI or distraction Norberg angle. 

The conclusion adds nothing to the literature.

Thank you for this comment. We added our major findings to the conclusion in order to gain more visibility in the literature. 

Line 334: This study confirms that long-term selection based on hip radiograph screening reduced the HD prevalence from 1997 to 2017 in the Cane Corso, Gordon Setter, Rottweiler and White Swiss Shepherd. It demonstrated that phenotypic selection for hip conformation may be effective, although it is dependent on the voluntary participation of breeders and owners. The true prevalence of HD in the breeds presented in this study is probably higher than those reported in our results. However, this screening type remains the only official procedure in most countries. To achieve a further decrease in the HD prevalence, communicating with veterinary practitioners and breeders on the value of classification is necessary in association with the use of EBV and genomic selection which should be considered. 

Reviewer #3: (changes requested by reviewer 2 are highlighted in green in the manuscript) 

This is a generally well written paper and the topic is interesting.

Nonetheless, I have a few concerns regarding the handling of the data:

1. The time frame of the observations vary greatly among breeds and, therefore, they are poorly comparable. Indeed, some breeds have data covering more than 30 ears and some others only 6 or 7 years. I see that the authors have chosen strict exclusion criteria to limit inter-observer variability in the evaluation of HD, and I also see that such a limitation is difficult to overcome. Therefore I think that the authors should discuss broadly such differences in the discussion.

Thank you for this constructive comment. Although we were not comparing HD prevalence between breeds, we agree that the division into several heterogenous cohorts is difficult to understand and may introduce bias. As a consequence, we will only keep, for the study, the 10 breeds which were divided in 3 homogeneous cohorts of 7 years from 1997 to 2017: the Alaskan Malamute, the Australian Shepherd, the Belgian shepherd dog, the Berger de Brie, the Cane Corso, the English Cocker Spaniel, the Gordon Setter, the Rottweiler, the Siberian Husky and the White Shepherd dog. 

We are also well aware that having all the radiographs analyzed by a single reader, which is the situation in several European countries, prevents any inter-observer variability (well described in the literature), but cannot prevent an intra-observer variability over the long period of time. This is why we wrote (lines 274-275) : yet an intraobserver variability over the long study period cannot be totally excluded which could introduce bias

2. At lines 267-276 the authors describe the variations of the different HD grades in time. I think that this part belongs to the results section. Moreover, I do not see any reference to such a comparison in the description of the statistical analysis. The authors should run appropriate statistical tests on this part (that is, in my opinion, the most interesting). A simple description of the trend in the graphs is not enough to make conclusions.

Thank you for this interesting comment. We modified the statistical analysis accordingly and placed the description into the results section (Line 147 to 175). 

Line 111: For each breed and each period, HD prevalence (expressed as %) was obtained by dividing the number of dogs that scored C-D and E by the total number of dogs evaluated for the breed. 

Within each breed, prevalences among A+B dogs and C+D+E dogs; and between A+B dogs and C, D and E dogs; for each period were compared using Fisher's exact test. Statistical analyses were performed by one author (AB) using a commercial software program (Prism 6, GraphPad Software, La Jolla, USA, CA).

3. The authors should find a way to summarize the graphical representation of the data. Presenting 28 different graph is very uncomfortable to read.

Thank you for this comment. In order to synthetize the data, we chose to present our results in 4 figures (one showing the breeds with a diminishing prevalence of HD, one showing the breeds with a diminishing and increasing prevalence of HD over the study period or increasing prevalence of HD. The 2 other figures present the same breeds but with the proportions of A-B grades and C-D-E grades over the study period). We think that the presentation of the results is therefore easier. 

Line 177 to 186: Figures 2 to 5.

---

## [Decision Letter · Decision Letter 1]

4 Jun 2020

PONE-D-20-10280R1

Prevalence of canine hip dysplasia in 10 breeds in France, a retrospective study of the 1997-2017 radiographic screening period

PLOS ONE

Dear Dr. Baldinger

Thank you for submitting your manuscript to PLOS ONE. After careful consideration, we feel that it has merit but does not fully meet PLOS ONE’s publication criteria as it currently stands. Therefore, we invite you to submit a revised version of the manuscript that addresses the points raised during the review process.

Many thanks for resubmitting your manuscript to PLOS One

It was reviewed by the same reviewers as last time, and one reviewer has recommended some other minor changes be made

If you could make these changes and write a response to reviewers, then the review can be expedited when resubmitted.

I wish you the best of luck with your revisions

Hope you are keeping safe and well in these difficult times

Thanks

Simon

We look forward to receiving your revised manuscript.

Kind regards,

Simon Clegg, PhD

Academic Editor

PLOS ONE

Reviewers' comments:

Reviewer's Responses to Questions

**Comments to the Author**

1. If the authors have adequately addressed your comments raised in a previous round of review and you feel that this manuscript is now acceptable for publication, you may indicate that here to bypass the “Comments to the Author” section, enter your conflict of interest statement in the “Confidential to Editor” section, and submit your "Accept" recommendation.

Reviewer #1: (No Response)

Reviewer #3: All comments have been addressed

2. Is the manuscript technically sound, and do the data support the conclusions?

Reviewer #1: Partly

Reviewer #3: Yes

3. Has the statistical analysis been performed appropriately and rigorously? 

Reviewer #1: Yes

Reviewer #3: Yes

4. Have the authors made all data underlying the findings in their manuscript fully available?

Reviewer #1: Yes

Reviewer #3: Yes

5. Is the manuscript presented in an intelligible fashion and written in standard English?

Reviewer #1: No

Reviewer #3: Yes

6. Review Comments to the Author

Reviewer #1: I think the structure and the work of content has greatly improved in this new version

Delete table 1 and Fig 1, 4 and 5 (information about HD prevalence not important, the table 2 and Fig. 2 and 3 contain this information separated by evaluation period)

Add a table with the number of dogs per breed and period and % of registered animals screened per period

Line 211-215 – The reduction of prevalence was between of 1 and 2 period between 2 and 3 the reduction was not significant; this should be not omitted and deserves an explanation.

Line 300-301- “D and E FCI scores at 12 months cannot be reliably predicted from the 4-month value of DI or distraction Norberg angle. I don`t know which was the reference but disagree with previous studies, the prediction of moderate and severe HD is reliably at 4 months of age using DI for passive hip laxity measurement.

There is some recent published works about CHD prevalence in other countries that should be added and results in terms of progress compared and discussed.

Reviewer #3: (No Response)

7. PLOS authors have the option to publish the peer review history of their article (what does this mean?). If published, this will include your full peer review and any attached files.

Reviewer #1: No

---

## [Author Response · Author response to Decision Letter 1]

16 Jun 2020

Response to reviewers and academic editor : 

We would like to thank the reviewers but also the academic editor for their comments and suggestions which contributed to our article improvement. 

Reviewer #1: (changes requested by reviewer 1 are highlighted in blue in the manuscript)

Dear Reviewer 1, 

We thank you for your interest in our manuscript. We appreciate your constructive comments. Please find hereunder the changes made to the manuscript following your comments. 

Delete table 1 and Fig 1, 4 and 5 (information about HD prevalence not important, the table 2 and Fig. 2 and 3 contain this information separated by evaluation period).

Table 1 and Fig 1,4 and 5 were deleted. 

Add a table with the number of dogs per breed and period and % of registered animals screened per period.

Thank you for this comment, we added the table as requested. However we were not able to provide the percentage of registered animals as this study was conducted with the data from the official lecturer of the presented breeds, the kennel club of each breed was not solicited for this study.

Breed 1997-2017 (N) 1997-2003 (N) 2004-2010 (N) 2011-2017 (N)

Cane Corso 1338 201 542 595

Gordon Setter 1803 900 594 309

White Swiss Shepherd dog 2924 225 1063 1636

Berger de Brie 1631 777 573 281

Rottweiler 7072 4539 1418 1115

English Cocker Spaniel 812 203 231 378

Australian Shepherd dog 4442 210 1469 2763

Alaskan Malamute 897 206 293 398

Belgian Shepherd dog 4998 1668 1796 1534

Siberian Husky 1870 380 397 1093

Table 2. Number of evaluated dogs (N) for the 10 breeds for each study period.

Line 136: The number of evaluated dogs for the 10 breeds is presented in Table 1. 

Line 211-215 – The reduction of prevalence was between of 1 and 2 period between 2 and 3 the reduction was not significant; this should be not omitted and deserves an explanation.

Thank you for this comment. The statistically significant decrease in HD prevalence was noted in these 4 breeds between period 1 and 2 but also between 1 and 3, therefore to be more specific we added the sentence “over the study period”. 

Line 203: A diminishing prevalence of HD was noted in 6 breeds in this study. Among them, 4 breeds (Cane Corso, Gordon Setter, Rottweiler and White Swiss Shepherd) showed a significant change in HD prevalence over the study period.

Line 216: A previous study demonstrated that when all dogs in a breed have nearly the same hip phenotype, almost no selection pressure can be applied to improve hip quality based on hip radiograph screening �14�. According to the results of the present study, this was potentially the case for the Siberian Husky and for the Australian Shepherd dog, which demonstrated slight changes in HD prevalence. This was also potentially the case for the Cane Corso, the Gordon Setter and the Rottweiler between the second and the third period of the study where the diminishing prevalence of HD was not statistically significant. 

Line 300-301- “D and E FCI scores at 12 months cannot be reliably predicted from the 4-month value of DI or distraction Norberg angle. I don`t know which was the reference but disagree with previous studies, the prediction of moderate and severe HD is reliably at 4 months of age using DI for passive hip laxity measurement.

Thank you for this comment. The reference was from Taroni et al. (VCOT, 2018), we removed this sentence from the manuscript. 

There is some recent published works about CHD prevalence in other countries that should be added and results in terms of progress compared and discussed.

Thank you for this comment. 

Several study from other countries were already added in this manuscript for comparison (Leighton et al. 2019, Oberbauer et al. 2017, Ohlerth et al. 2019, Wang et al. 2019, James et al. 2020, Hedhammar et al. 2020). Due to the large number of references about canine hip dysplasia, this review references were limited to the most appropriate to our study. We re-evaluated the recent bibliography and found two studies which were not included in our study (Kirberger et al. 2017; Wilson et al. 2015). We added the references in our work.

---

## [Editor Report · Decision Letter 2]

24 Jun 2020

Prevalence of canine hip dysplasia in 10 breeds in France, a retrospective study of the 1997-2017 radiographic screening period

PONE-D-20-10280R2

Dear Dr. Baldinger

We’re pleased to inform you that your manuscript has been judged scientifically suitable for publication and will be formally accepted for publication once it meets all outstanding technical requirements.

Kind regards,

Simon Clegg, PhD

Academic Editor

PLOS ONE

Additional Editor Comments:

Many thanks for re-submitting your manuscript to PLOS One

I have reviewed the manuscript, and it reads well, and as you have addressed all the reviewer comments, I have recommended the article for publication

You should hear from the Editorial Office soon

It was a pleasure working with you, and I wish you all the best for your future research

Hope you are keeping safe and well in these difficult times

Thanks

Simon

---

## [Editor Report · Acceptance letter]

26 Jun 2020

PONE-D-20-10280R2 

Prevalence of canine hip dysplasia in 10 breeds in France, a retrospective study of the 1997-2017 radiographic screening period 

Dear Dr. Baldinger:

I'm pleased to inform you that your manuscript has been deemed suitable for publication in PLOS ONE. Congratulations! Your manuscript is now with our production department. 

Kind regards, 

on behalf of

Dr. Simon Clegg 

Academic Editor

PLOS ONE